# Impact of COVID-19 on Lung Disease in People with Cystic Fibrosis: A 6-Month Follow-Up Study on Respiratory Outcomes

**DOI:** 10.3390/biomedicines10112771

**Published:** 2022-11-01

**Authors:** Paola Medino, Gianfranco Alicandro, Chiara Rosazza, Fabiana Ciciriello, Andrea Gramegna, Arianna Biffi, Valeria Daccò, Vincenzina Lucidi, Marco Cipolli, Mariaserena Boraso, Erica Nazzari, Carla Colombo

**Affiliations:** 1Cystic Fibrosis Centre, Fondazione IRCCS Ca’ Granda Ospedale Maggiore Policlinico, Via della Commenda 9, 20122 Milano, MI, Italy; 2Department of Pathophysiology and Transplantation, University of Milan, Via Francesco Sforza 35, 20122 Milano, MI, Italy; 3Cystic Fibrosis Centre, Bambino Gesù Children’s Hospital, IRCCS, Piazza S. Onofrio 4, 00165 Roma, RM, Italy; 4Internal Medicine Department, Respiratory Unit and Regional Adult Cystic Fibrosis Centre, Fondazione, IRCCS Cà Granda Ospedale Maggiore Policlinico, Via Francesco Sforza 35, 20122 Milano, MI, Italy; 5Cystic Fibrosis Centre, Azienda Ospedaliero Universitaria Integrata di Verona, Piazzale Stefani 1, 37126 Verona, VR, Italy

**Keywords:** COVID-19, cystic fibrosis, respiratory outcomes, ppFEV1, pulmonary exacerbation

## Abstract

Background: The impact of COVID-19 on respiratory outcomes in people with cystic fibrosis (pwCF) has not been clearly characterized. We evaluated changes in respiratory function indicators derived from spirometry and pulmonary exacerbation rates 6 months after SARS-CoV-2 infection. Methods: This multicentre prospective study was based on pwCF enrolled between October, 2020 and June, 2021 in the DECO COVID-19 project. PwCF complaining of COVID-like symptoms were tested with real-time polymerase chain reaction (RT-PCR) for SARS-CoV-2 on nasopharyngeal swab. Mean changes in respiratory function indicators and time to first episode of pulmonary exacerbation were compared between RT-PCR-positive and RT-PCR-negative patients. Regression models were used to adjust for baseline percent predicted forced expiratory volume in one second (ppFEV1) values, number of comorbidities, and initiation of CFTR modulator therapy during the follow-up. Results: We enrolled 26 pwCF with RT-PCR-confirmed infection and 42 with a RT-PCR-negative test. After 6 months of follow-up, mean ppFEV1 changes were not significantly different between groups (+0.3% in positive vs. +0.2% in negative patients, *p* = 0.19). The 6-month cumulative probabilities of a first episode of pulmonary exacerbation were: 0.575 among RT-PCR-negative patients and 0.538 among those with a positive test (adjusted hazard ratio: 0.88, 95% CI: 0.44–1.75). Conclusions: COVID-19 did not appear to negatively affect respiratory outcomes of pwCF at 6 months from infection.

## 1. Introduction

Cystic fibrosis (CF) is the most frequent autosomal recessive genetic disease among Caucasians. It is a potentially life-threatening multisystem condition caused by mutations of the CF transmembrane regulator (CFTR) gene, which encodes a chloride ion channel embedded in the apical membrane of epithelial cells, where it regulates ionic transport and fluid homeostasis. Its alteration causes the production of thick secretions with consequent obstruction of glandular ducts and progressive organ damage [1].

The main clinical manifestations of CF are related to pancreatic insufficiency and lung disease with fat malabsorption and recurrent respiratory infections. Lung disease remains the major cause of morbidity and mortality in people with CF (pwCF).

Over the last decades, remarkable advances have been made in the management of the disease with half of pwCF expected to reach their fifties [2], and further improvements in survival are expected to occur following the introduction of highly effective CFTR modulators [1,3].

Viral infections in pwCF frequently trigger respiratory exacerbations and contribute to the deterioration of respiratory function [4]. In 2009, H1N1 flu caused significant morbidity in pwCF and, in those with underlying severe lung disease, it was associated with respiratory deterioration, need of mechanical ventilation, and even death [5,6,7]. Thus, when the novel coronavirus disease-19 (COVID-19) broke out, pwCF were considered at high risk for severe disease.

However, early studies at the start of the pandemic did not report COVID-19 to be more severe in pwCF than in the general population [8,9,10,11,12]. Later observations on larger numbers of patients revealed that pwCF with severe pulmonary disease, CF-related diabetes, or organ transplantation are at high risk of severe COVID-19 [11,12,13,14,15]. However, at present, it is still unclear whether COVID-19 may cause chronic and irreversible lung damage and favour the progression of CF-related pulmonary disease [10].

In this context, our study aims to evaluate the clinical manifestations of COVID-19 in pwCF and to ascertain its long-term impact on two major CF respiratory outcomes, lung function and pulmonary exacerbation (PE).

## 2. Materials and Methods

The study was based on pwCF enrolled in the DECO COVID-19 project, a multicentre prospective study supported by the Italian Ministry of Health (COVID-2020-12371781), that involved three Italian Regional Reference Centres for CF (Milan, Rome, and Verona). Approvals by the Ethics Committee of the Fondazione IRCC Ca’ Granda Ospedale Maggiore Policlinico, Milan (Italy) (coordinating centre) and by the ethics committees of the other institutions involved in our multicentre study were obtained and the informed consent was signed before patients’ inclusions in the study.

Between 15 October 2020 and 30 June 2021, all patients consecutively attending the CF centres with COVID-like symptoms were enrolled in this study. At enrolment, patients underwent a nasopharyngeal swab and tested for presence of SARS-CoV-2 by real-time polymerase chain reaction (RT-PCR). Demographic and clinical data were then collected, including sex, age, CFTR genotype, body mass index (BMI), maintenance therapy, pre-existing CF-related comorbidities, transplant status, respiratory microbiology, and average value of respiratory function indicators obtained from the last three spirometries. In addition, need of hospitalization, admission to the intensive care unit (ICU), need of oxygen supplementation, non-invasive and invasive ventilation, and radiological data were collected. Diagnosis of PE was made according to the following clinical features: increased cough, increased sputum production, shortness of breath, chest pain, loss of appetite, weight loss, and decrease in spirometric parameters [16].

Follow-up outcomes considered after 6 months from the RT-PCR test included vital status, respiratory function indicators, and time to first PE. Forced expiratory volume in one second (FEV1), forced vital capacity (FVC), and FEV1/FVC ratio, were expressed as percent of predicted values using the Global Lung Initiative reference values [17].

Data were collected using REDCap (12.0.19—© 2022 Vanderbilt University), an electronic data capture tool hosted at the Fondazione IRCCS Ca’ Granda Ospedale Maggiore Policlinico of Milan.

Primary outcomes included changes in respiratory function indicators after 6 months from RT-PCR and time to first PE over the same period. We also evaluated chest radiological findings and sputum microbiology.

Mean differences in absolute changes (from enrollment to 6-month follow-up visit) in respiratory function indicators between patients who tested positive and those who tested negative and their corresponding 95% confidence interval (CI) were estimated using linear regression models. The cumulative incidence of first PE over the 6-month follow-up was computed using the Kaplan–Meier technique, and differences between groups were evaluated using the log-rank test. The hazard ratio (HR) of PE was also estimated through a Cox regression model. All regression models were adjusted for age, number of comorbidities, and introduction of CFTR modulator therapy during the follow-up (Yes/No). The models for respiratory function indicators were further adjusted for baseline values.

All statistical tests were performed at 0.05 significance level.

## 3. Results

We enrolled 68 patients with acute respiratory symptoms, including 26 pwCF with RT-PCR-confirmed COVID-19 (median age: 29 years, range: 1–66) and 42 pwCF who tested negative to RT-PCR and received a diagnosis of CF pulmonary exacerbation (median age: 26 years, range: 3 months-51). For 15 of the 26 patients with positive RT-PCR test we identified the SARS-CoV-2 lineage: 8 patients were infected by B.1.1.7 (Alpha variant), 3 by B.1.177, 2 by AY.43 (Delta variant), 1 by AY.36 (Delta variant), and 1 by B.1.636.

Table 1 reports the baseline characteristics of the two groups of patients, which were comparable for sex, age, CFTR genotype, pancreatic status, respiratory microbiology, maintenance therapy, and presence of comorbidities. Baseline ppFEV1 and ppFVC values were lower in the RT-PCR negative group, whereas ppFEV1/FVC values were comparable.

In both groups, the most frequent symptoms were cough, fever, and dyspnea. In the COVID-19 group, systemic symptoms such as fever, myalgia, and asthenia were more frequent, as well as upper respiratory tract symptoms and headache. Diarrhoea, pharyngodinia, and anosmia were reported only by patients with COVID-19 (Table 2). Persistence of symptoms at 12 weeks was observed in one patient in the COVID-19 group and in one in the negative group. Ten patients in the RT-PCR positive group (38.5%) were hospitalized for COVID-19, while 38 patients who tested negative (90.5%) needed hospitalization, mostly for intravenous antibiotic therapy. In fact, almost all patients with a RT-PCR negative test received IV antibiotics, according to the bacteria identified in the sputum culture, whereas only 7% received IV antibiotics, and around 50% received oral antibiotic therapy in the RT-PCR positive group. Antiviral therapy was administered in about 15% of the positive patients. Median length of hospital stay (interquartile range) was not significantly different between groups: 15 days (13, 17) among RT-PCR negative and 16 days (14, 35) among RT-PCR positive patients (*p* = 0.41). One patient in the positive group needed sub-intensive care but none were admitted to the ICU, while this occurred in one negative patient. Need of oxygen supplementation was similar in the two groups (11.5% in the RT-PCR positive vs. 11.9% in the RT-PCR negative). All subjects fully recovered without sequelae.

Radiological evaluation could be performed mostly in patients who required hospitalization. Overall, 11 RT-PCR-positive and 29 RT-PCR-negative patients underwent chest X-ray, while 7 RT-PCR-positive and 20 RT-PCR-negative patients received chest CT examination. Lobar consolidation was less frequently observed among patients with COVID-19 as compared to RT-PCR-negative subjects [2 (18.2%) vs. 17 (58.6%), *p* = 0.022], while interstitial pneumonia occurred more frequently in COVID-19 patients [9 (81.8%) vs. 12 (41.4%), *p* = 0.022] (Appendix A). Figure 1 illustrates radiological images (chest X-ray and CT) obtained from the same patient during the acute phase of COVID-19 and after 6 months of follow-up showing a complete recovery of the typical radiological COVID-19 findings. Six out of nine positive subjects with interstitial pneumonia had a radiological follow-up available and only one of them had a chest X-ray showing persistence of bilateral patchy reticular opacities at 6-month follow-up.

Six months after enrollment, respiratory function indicators had not significantly worsened in both groups (Table 3). Over the same period, PE occurred in 23/40 RT-PCR negative patients (follow-up data not available in 2 patients) and in 14/26 RT-PCR-positive subjects. Figure 2 shows the cumulative probability of PE among RT-PCR-positive patients and among those with a negative test. The 6-month cumulative probabilities were 0.538 (95% CI: 0.301–0.695) in RT-PCR-positive patients and 0.575 (95% CI: 0.391–0.704) in RT-PCR-negative patients. No significant differences emerged between groups (HR: 0.86, 95% CI: 0.44–1.68, adjusted HR: 0.88, 95% CI: 0.44–1.75).

Similarly, no significant differences emerged in prevalence of positive sputum culture for *P. aeruginosa* at 6-month follow-up in the two groups (Appendix A).

## 4. Discussion

In this study, we found that COVID-19 did not impact respiratory outcomes in CF patients, including pulmonary function and time to first episode of PE at 6 months from infection. This favorable evolution occurred despite the high rate of radiological evidence of interstitial pneumonia in those who developed COVID-19.

In a previous Italian multicentre study, we found no significant reduction in ppFEV1 values in 236 pwCF with COVID-19 after a median time from diagnosis of 2 months [13]. The present study provides further information on the consequences of COVID-19, although in a lower number of pwCF, by extending the period of follow-up from 2 to 6 months from its diagnosis and also reporting on radiological evolution of the disease.

Data so far available suggest that pwCF, unlike what would be expected, do not experience a particularly severe course of the disease, as indicated by the low rate of patients needing oxygen support and ICU admission as well as by the low case fatality rate (<3%) [9,10,11,12,13,15,18]. Only Hadi et al., in the US, found a worse outcome in pwCF after SARS-CoV-2 infections as compared to a group of patients without CF matched through a propensity score based on sex, age, race, BMI, smoking, and comorbidities. Data from that retrospective study reported higher rates of hospitalization, critical care need, acute renal injury and 30-day mortality in pwCF as compared to patients without CF (5.3% vs. 2.9%) [19]. However, the median age of pwCF included in that study was very high (47 years), the highest ever reported in studies on pwCF and higher than the median age at death of this population. Another limit is the lack of control for lung transplantation in the analysis, a condition that is more frequent in pwCF than in the general population and represents the strongest risk factor for hospitalization, need of oxygen therapy, and death from COVID-19 [15,18]. Data published by Naehrlich et al. documented that lung-transplanted pwCF with COVID-19 are 1.7-fold more frequently admitted to hospital and require 8-fold more frequent oxygen support than those without lung transplant [9].

Thus, although data on the severity of COVID-19 in pwCF are not consistent among different studies, most of them suggest that the clinical course is relatively mild in the absence of a few risk factors, including severe pulmonary disease, CF-related diabetes, and organ transplantation.

Our data suggest that the clinical course of COVID-19 is not more severe than the common clinical course of a respiratory exacerbation. Furthermore, symptomatic infections for SARS-CoV-2 did not change baseline spirometric values after 6 months of follow-up and did not increase the risk of PE in the 6 months following infection.

Even though differential diagnosis may be difficult, we found some clinical features of SARS-CoV-2 infection, such as the high frequency of systemic symptoms and upper respiratory tract involvement, which may help clinicians discerning a symptomatic SARS-CoV-2 infection from an episode of pulmonary exacerbation. In agreement with Corvol et al. [15], we found that the typical clinical features of COVID-19 (e.g., fever, dry cough, myalgia) in pwCF appeared to be similar to those observed in the general population with COVID-19.

In addition, persistence of symptoms at 12 weeks from infection, reported in up to 40% of infected individuals without CF [20], was rarely observed in pwCF, as a likely consequence of their younger age. Our findings refer mostly to the second wave of the pandemic in Italy, when the Alpha variant was the predominant one.

Especially in the first wave of the pandemic, lower SARS-CoV-2 infection rates have been reported in pwCF as compared to the general population [8,9,14,21,22,23]. In a national study involving all French CF centres from March 2020 to April 2021, Corvol et al. reported a low cumulative incidence of COVID-19 (3%) among the overall population, but a higher risk of SARS-CoV-2 infection in post-transplant and older individuals [15].

Similarly, in a multicentre Italian study, 236 out of 4300 pwCF with positive RT-PCR test results for SARS-CoV-2 were referred by CF Centres between March 2020 and June 2021, with a cumulative incidence of 5.5% [13].

Possible factors that may explain the lower risk of infection include regular use of preventive measures, such as face masks and hand sanitization, as well as a protective role of airway clearance, mucolytics (dornase alfa), and antibiotic prophylaxis with azithromycin that may exert an anti-inflammatory effect [8]. In addition, the peculiarity of the CF lung environment and SARS-CoV-2 infection susceptibility of CF airway epithelial cells may play a role. CF airway epithelial cells may be less susceptible to SARS-CoV-2 infection due to reduced entry of the virus and to altered intracellular processes involved in host defense and viral replication [24,25,26]. Suryamohan et al. hypothesized that mutations in the CFTR gene may alter the expression of SARS-CoV-2 receptors (ACE2) and co-receptors (TMPRSS2), thereby mitigating the effects of SARS-CoV-2 infection and lung damage in pwCF [26]. We may also speculate that the altered properties of the abundant mucins present in CF airways may represent an additional barrier to SARS-CoV-2 entry and spreading.

This study had some limitations, including the relatively small sample size that did not allow separate evaluation of highly-vulnerable patients, such as those with severe lung disease and transplant recipients. In addition, it should be considered that when the study started COVID-19 vaccines were not yet available.

Despite these limitations, to our knowledge, this is the first study that evaluated respiratory outcomes 6 months after diagnosis of COVID-19 in pwCF. Our data suggest that generally, COVID-19 does not increase the progression of CF-related pulmonary disease. These findings are based on the two major outcome measures for morbidity and mortality in CF, i.e., ppFEV1 and PE, and on a comparison with a RT-PCR-negative control group tested for COVID-like symptoms.

Furthermore, the study included both paediatric and adult pwCF, with an age distribution which mirrors that of the Italian patient population [27], thus suggesting a high generalizability of our results.

In conclusion, our data indicate that COVID-19 does not seem to have a negative impact on respiratory outcomes 6 months after infection. Longer follow-up is needed to monitor possible longer-term effects of COVID-19.

## Figures and Tables

**Figure 1 biomedicines-10-02771-f001:**
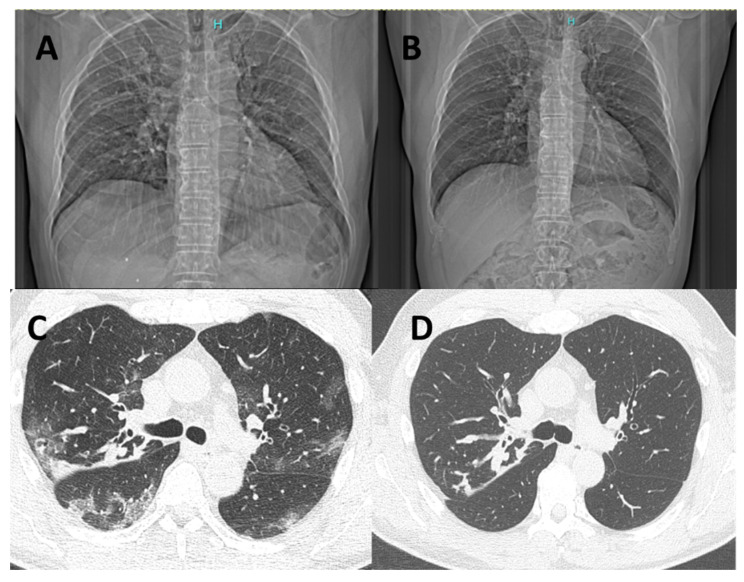
Chest X-ray and chest CT of a patient affected by COVID-19 in the acute phase of the disease (**left**, **A**,**C**) and after 6 months of follow-up (**right**, **B**,**D**). Chest CT shows radiological features of increased density and ground-glass opacities with peripheral and confluent distribution (**C**). The follow-up scan at 6 months documents complete resolution of COVID-related radiological findings, while peribronchial thickening and CF-related bronchiectasis remained unchanged (**D**).

**Figure 2 biomedicines-10-02771-f002:**
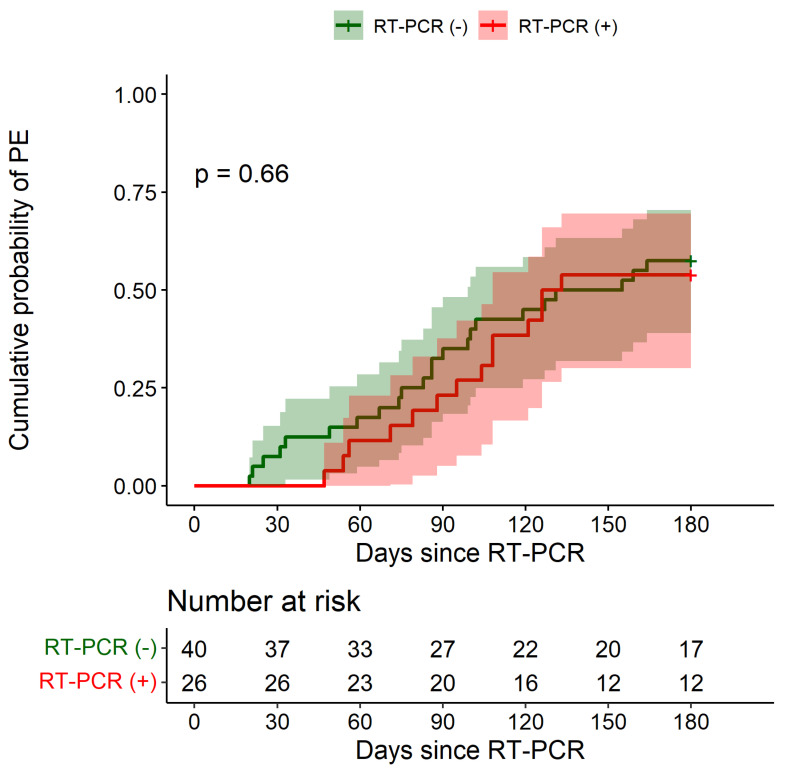
Cumulative probability of first pulmonary exacerbation over the 6-month period after RT-PCR test according to SARS-CoV-2 infection status. PE: pulmonary exacerbation. Shaded areas indicate 95% confidence intervals.

**Table 1 biomedicines-10-02771-t001:** Characteristics of the study population according to SARS-CoV-2 infection status.

	RT-PCR (−)		RT-PCR (+)		*p* Value ^a^
	No.	%	No.	%	
Tot.	42	100	26	100	
** *Sex* **					0.095
Males	14	33.3	14	53.8	
Females	28	66.7	12	46.2	
***Age group*** (*years*)					0.077
<18	8	19	7	26.9	
18–39	30	71.4	12	46.2	
40+	4	9.5	7	26.9	
** *CFTR genotype* **					0.29
F508del homozygous	6	14.3	7	26.9	
F508del heterozygous	18	42.9	12	46.2	
Other mutations	18	42.9	7	26.9	
** *Pancreatic insufficiency* **	32	76.2	17	65.4	0.33
** *ppFEV1, mean (SD)* **	68.5	24.3	82.5	24.9	0.032
** *ppFVC, mean (SD)* **	81.3	22.7	92.3	17.8	0.035
** *ppFEV1/FVC, mean (SD)* **	80.7	16.6	82.7	16.8	0.67
** *Respiratory microbiology* **					
*P. aeruginosa infection*	31	73.8	16	61.5	0.43
*A. xylosoxidans*	7	16.7	4	15.4	1.00
*S. maltophilia*	7	16.7	2	7.7	0.47
MRSA	6	14.3	6	23.1	0.51
*B. cepacia* complex	1	2.4	0	0	-
NTM	2	4.8	2	7.7	0.63
*Aspergillus* spp.	17	40.5	4	15.4	0.057
** *Lung transplantation* **	5	11.9	1	3.8	0.39
** *Comorbidities* **					
Diabetes	11	26.2	6	23.1	0.77
Liver disease	15	35.7	6	23.1	0.27
Hypertension	3	7.1	5	19.2	0.24
CVD	2	4.8	1	3.8	1.00
Renal disease	5	11.9	1	3.8	0.39
Cancer	0	0	0	0	-
Number of comorbidities					0.64
None	20	47.6	15	57.7	
At least one	12	28.6	5	19.2	
Two or more	10	23.8	6	23.1	
** *Maintenance therapy* **					
Oxygen	4	9.5	0	0	0.29
Inhaled antibiotics	16	38.1	12	46.2	0.51
Systemic antibiotics	16	38.1	5	19.2	0.10
Azithromycin	20	47.6	9	34.6	0.32
Inhaled steroids	32	76.2	15	57.7	0.11
Systemic steroids	6	14.3	2	7.7	0.70
CFTR modulators	7	16.7	6	23.1	0.54
Immunosuppressive drugs	4	9.5	1	3.8	0.64
Anti-hypertensive drugs	3	7.1	5	19.2	0.24

Data are numbers and percentage, unless otherwise specified. CVD: cardiovascular diseases. MRSA: methicillin-resistant Staphylococcus aureus. NTM: nontuberculous mycobacteria. ppFEV: forced expiratory volume in one second, expressed as percent of predictive value. ppFVC: forced vital capacity, expressed as percent of predicted. ppFEV1/FVC: FEV1/FVC ratio expressed as percent of predicted. RT-PCR: reverse transcriptase-polymerase chain reaction. ^a^: chi-square test or Fisher exact test (when 50% of the cells have expected counts less than 5) for comparison of categorical variables between patients with RT-PCR positive vs. negative result. *t*-test for independent samples for comparison of respiratory function indicators between groups.

**Table 2 biomedicines-10-02771-t002:** COVID-19-related symptoms, treatment, and severity according to SARS-CoV-2 infection status.

	RT-PCR (−)		RT-PCR (+)		*p*-Value ^a^
	No.	%	No.	%	
Tot.	42	100	26	100	
*Symptoms*					
Fever	18	42.9	18	69.2	0.034
Headache	2	4.8	6	23.1	0.047
Joint pain	2	4.8	5	19.2	0.097
Myalgia	0	0	6	23.1	0.002
Asthenia	8	19	16	61.5	0.001
Dyspnea	17	40.5	8	30.8	0.42
Chest pain	4	9.5	3	11.5	1.00
Cough	30	71.4	16	61.5	0.40
Rhinitis	2	4.8	6	23.1	0.047
Pharyngodinia	0	0	7	26.9	0.001
Anosmia	0	0	4	15.4	0.018
Abdominal pain	2	4.8	3	11.5	0.36
Diarrhoea	0	0	5	19.2	0.006
Vomiting	2	4.8	1	3.8	1.00
Persistence of symptoms at 12 weeks	1	2.4	1	3.8	-
*Treatment for COVID-19*					
Antiviral therapy	0	0	4	15.4	<0.001
Intravenous antibiotics	39	92.9	11	42.3	<0.001
Oral antibiotics	3	7.1	12	46.2	<0.001
Inhaled steroids	6	14.3	2	7.7	0.70
Systemic steroids	12	28.6	12	46.2	0.19
NSAID	0	0	1	3.8	0.38
Intravenous immunoglobulins	0	0	0	0	-
Monoclonal antibodies	0	0	0	0	-
Heparin	2	4.8	8	30.8	0.005
*Hospitalization*	38	90.5	10	38.5	<0.001
*Hospitalization unit*					-
Non-intensive care	37	88.1	9	34.6	
Sub-intensive unit	0	0	1	3.8	
Intensive unit	1	2.4	0	0	
*Needing oxygen support*					1.00
No	35	83.3	22	84.6	
Yes	5	11.9	3	11.5	
Already in oxygen support/no increase in dependency	2	4.8	1	3.8	
*Mechanical ventilation*	1	2.4	1	3.8	1.00
*CPAP*	1	2.4	3	11.5	0.15
*HFNC*	3	7.1	1	3.8	1.00
*ECMO*	1	2.4	1	3.8	1.00
*Clinically recovered*	42	100	26	100	-

CPAP: continuous positive airway pressure. ECMO: extracorporeal membrane oxygenation. HFNC: high-flow nasal cannula. NSAID: non-steroidal anti-inflammatory drugs. RT-PCR: reverse transcriptase-polymerase chain reaction. ^a^: chi-square test or Fisher exact test (when 50% of the cells have expected counts less than 5) for comparison between patients with RT-PCR positive vs. negative result.

**Table 3 biomedicines-10-02771-t003:** Changes in pulmonary function indicators from baseline to 6 months after the RT-PCR test according to SARS-CoV-2 infection status.

	Non-Missing Values	Baseline, Mean (SD)	6 Months, Mean (SD)	Mean Change (95% CI)	Between-Group Difference in Change (95% CI) ^a^	Adjusted between-Group Difference in Change (95% CI) ^a^
* **ppFEV1** *						
RT-PCR (+)	24	82.5 (24.9)	82.8 (23.8)	0.3 (−3.3, 3.9)	0.2 (−6.6, 6.9) *p* = 0.97	3.8 (−1.9, 9.4) *p* = 0.19
RT-PCR (−)	38	69.7 (23.8)	69.9 (24.2)	0.2 (−5.0, 5.3)		
* **ppFVC** *						
RT-PCR (+)	24	92.3 (17.8)	93.5 (16.0)	1.2 (−2.9, 5.3)	−1.6 (−86, 5.4) *p* = 0.65	0.2 (−5.7, 6.0) *p* = 0.96
RT-PCR (−)	37	81.8 (22.4)	84.6 (25.9)	2.8 (−2.5, 8.2)		
* **ppFEV1/FVC** *						
RT-PCR (+)	20	82.7 (16.8)	86.5 (46.6)	3.8 (0.3, 7.2)	4.3 (−1.4, 9.9) *p* = 0.14	5.5 (0.8, 10.1) *p* = 0.021
RT-PCR (−)	36	81.9 (16.6)	81.4 (12.8)	−0.5 (−4.6, 3.5)		

ppFEV: forced expiratory volume in one second expressed as percent of predicted. ppFVC: forced vital capacity expressed as percent of predicted. RT-PCR: reverse transcriptase-polymerase chain reaction. ^a^: Between-group differences in changes and corresponding 95% confidence interval (bars) were estimated using linear regression models. The regression model was adjusted for baseline respiratory function value, age, number of comorbidities, and introduction of CFTR modulator therapy during the follow-up (Yes/No). One patient received lung transplantation during the follow-up and was not included in this analysis.

## Data Availability

The data presented in this study are available on request from the corresponding author. The data are not publicly available due to privacy and ethical restrictions.

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
