# Peer review of "Impact of COVID-19 on Lung Disease in People with Cystic Fibrosis: A 6-Month Follow-Up Study on Respiratory Outcomes"

_biomedicines, 2022, doi:10.3390/biomedicines10112771_

Round 1
Reviewer 1 Report (Previous Reviewer 2)
The authors have provided relevant answers to the reviewer comments. I still think the study's compasition of covid positive and negative CF patients is of limited interest, but it is soundly presented and scientifically ok.
Reviewer 2 Report (Previous Reviewer 1)
The authors have done great modifications.
This manuscript is a resubmission of an earlier submission. The following is a list of the peer review reports and author responses from that submission.
Round 1
Reviewer 1 Report
This study focuses on “Impact of COVID-19 on lung disease in people with cystic fibrosis: a 6-month follow-up study on respiratory outcomes” from Italy. The idea is interesting and important as there is limited detailed data on cystic fibrosis populations. However, there are incomplete data and some important concerns.
1. The authors should provide the details of covid variant in this study.
2. The authors should provide the details of covid treatment in the cystic fibrosis populations
3. The author should also define the term exacerbation with reference.
4. Cystic fibrosis (CF) score should be provided to show the severity at baseline
5. The lung function value and %predicted of FEV1, FVC, FEV1/FVC at baseline and 6-month follow-up should be provided.
6. How many people have received CT images at baseline and 6-month follow-up ? please add covid ct severity scores at baseline and 6-month follow-up.
Radiology. 2021 Apr;299(1):E177-E186.
7. How many people received lung transplant in this study? There is evidence that some subsets within the CF population, including those post-transplantation, may experience a more severe clinical course.
8. As for exacerbation in 6-month follow-up, please provide analysis of time to first exacerbation (KM curve).
9. The percentage of missing/not available data of clinical characters (e.g. CT, FEV1, ) should be provided.
10. The days of hospitalization should be provided.
11. The sputum microbiology at baseline and 6-month follow-up should be provided.
12. Did these patients receive 6-minute walk test at baseline and 6-month follow-up? provide data if available
Reviewer 2 Report
This manuscript reports findings from a study in pwCF comparing change in ppFEV1 and number of pulmonary exacerbations between those with a positive vs negative covid-19 test. A relatively small sample of participants (n=26 with covid-19, n=42 without) were recruited from 3 Italian CF centres after presenting with covid-19 like symptoms and followed up for 6 months.
Although long-term impact of covid-19 on respiratory outcomes in CF is a relevant topic, it is unclear how the comparison between covid-19 and other-cause-exacerbations in CF contribute to the existing knowledge. Specifically, more information about the comparison group is needed for the reader to make valid inferences from the findings. Which viral or bacterial infections were identified in the comparison group?
Also, it is unclear why the authors did not have access to a larger sample. They refer to their own previous studies based on 2 months follow-up of 236 participants with CF and covid-19. It would have been more interesting to report additional follow-up data from this larger cohort, especially since the larger sample showed relevant subgroup findings (more severe covid-19 in CF patients with low ppFEV1 and BMI) which are not possible to assess in the present small sample.
The authors’ conclusion that covid-19 did not have a negative impact on respiratory outcomes in CF at 6 months follow-up is not supported by the data, unless it is specified that covid-19 did not have worse negative impact compared to other causes of pulmonary exacerbations. For this finding to be relevant for clinicians and researchers, additional information is needed to describe these other causes. However, it would seem more relevant to compare pwCF with covid-19 to “healthy pwCF” i.e. without respiratory symptoms. If these data are available, we suggest that the authors make this comparison and preferably in a larger sample to enable subgroup analyses.